Bacterial image analysis using multi-task deep learning approaches for clinical microscopy

Chin Shuang Yee 1
Dong Jian dongjian@cesi.cn 2
Hasikin Khairunnisa khairunnisa@um.edu.my 1 3
Ngui Romano 4
Lai Khin Wee 1
Yeoh Pauline Shan Qing 1
Wu Xiang 1 5
1 Department of Biomedical Engineering, Faculty of Engineering, Universiti Malaya , Kuala Lumpur , Malaysia
2 China Electronics Standardization Institute , Beijing , China
3 Centre of Intelligent Systems for Emerging Technology (CISET), Faculty of Engineering, Universiti Malaya , Kuala Lumpur , Malaysia
4 Malaria Research Centre, Faculty of Medicine and Health Sciences, Universiti Malaysia Sarawak , Kota Samarahan , Sarawak , Malaysia
5 Institute of Medical Information Security, Xuzhou Medical University , Xuzhou , China
Alatas Bilal
Electronic publication date: 2024 Aug 8
Publication date: 2024
Volume: 10
Electronic Location ID: e2180
Received 2024 Feb 14; Accepted 2024 Jun 17
Copyright: ©2024 Chin et al.
Copyright year: 2024
Copyright holder: Chin et al.
License: This is an open access article distributed under the terms of the Creative Commons Attribution License, which permits unrestricted use, distribution, reproduction and adaptation in any medium and for any purpose provided that it is properly attributed. For attribution, the original author(s), title, publication source (PeerJ Computer Science) and either DOI or URL of the article must be cited.
License URL: https://creativecommons.org/licenses/by/4.0/

Keywords: Bacteria detection, Bacteria classification, Deep learning, Object detection, YOLOv4, EfficientDet, SSD-MobileNetV2, Microscopic images, Image analysis

Funding: National Key Research and Development Program of China No. 2020YFC2006600 This research is funded by the National Key Research and Development Program of China under Grant No. 2020YFC2006600. The funders had no role in study design, data collection and analysis, decision to publish, or preparation of the manuscript.

==============================
Background

Bacterial image analysis plays a vital role in various fields, providing valuable information and insights for studying bacterial structural biology, diagnosing and treating infectious diseases caused by pathogenic bacteria, discovering and developing drugs that can combat bacterial infections, etc. As a result, it has prompted efforts to automate bacterial image analysis tasks. By automating analysis tasks and leveraging more advanced computational techniques, such as deep learning (DL) algorithms, bacterial image analysis can contribute to rapid, more accurate, efficient, reliable, and standardised analysis, leading to enhanced understanding, diagnosis, and control of bacterial-related phenomena.

Methods

Three object detection networks of DL algorithms, namely SSD-MobileNetV2, EfficientDet, and YOLOv4, were developed to automatically detect Escherichia coli (E. coli) bacteria from microscopic images. The multi-task DL framework is developed to classify the bacteria according to their respective growth stages, which include rod-shaped cells, dividing cells, and microcolonies. Data preprocessing steps were carried out before training the object detection models, including image augmentation, image annotation, and data splitting. The performance of the DL techniques is evaluated using the quantitative assessment method based on mean average precision (mAP), precision, recall, and F1-score. The performance metrics of the models were compared and analysed. The best DL model was then selected to perform multi-task object detections in identifying rod-shaped cells, dividing cells, and microcolonies.

Results

The output of the test images generated from the three proposed DL models displayed high detection accuracy, with YOLOv4 achieving the highest confidence score range of detection and being able to create different coloured bounding boxes for different growth stages of E. coli bacteria. In terms of statistical analysis, among the three proposed models, YOLOv4 demonstrates superior performance, achieving the highest mAP of 98% with the highest precision, recall, and F1-score of 86%, 97%, and 91%, respectively.

Conclusions

This study has demonstrated the effectiveness, potential, and applicability of DL approaches in multi-task bacterial image analysis, focusing on automating the detection and classification of bacteria from microscopic images. The proposed models can output images with bounding boxes surrounding each detected E. coli bacteria, labelled with their growth stage and confidence level of detection. All proposed object detection models have achieved promising results, with YOLOv4 outperforming the other models.

Introduction

Microorganisms, including unicellular and multicellular organisms, provide vital functions in the natural environment and human existence. These microorganisms, such as bacteria, fungi, algae, viruses, and protozoa, can provide advantages. In contrast, a significant number of microorganisms can provide risks by causing various diseases, including but not limited to typhoid and cancer. On the one hand, specific microbes play an essential role in several processes, including fermentation in food production, sewage treatment, soil fertility management in agriculture, and drug formulations in medicine. Conversely, some microbes present substantial risks, giving rise to various diseases that span from commonplace infections to grave afflictions, such as acquired immunodeficiency syndrome (AIDS) and specific forms of cancer. Microorganisms are generally the species that play the most crucial role in the Earth’s ecosystem. However, humans must be aware that microorganisms pose a significant threat to other organisms (Kotwal et al., 2022; Zhang et al., 2021).

Bacteria are tiny, primitive, unicellular, prokaryotic, and microscopic organisms with simple structures and without nucleus cytoskeletons, as well as membranous organelles. They range in size from less than ten micrometres (µm); therefore, they cannot be observed with the naked eye and must be studied with a microscope. Bacteria are classified into different groups based on their morphology, which refers to their shape. This classification includes spherical bacteria known as cocci, spiral-shaped bacteria known as spirilla, rod-shaped bacteria known as bacilli, comma-shaped bacteria referred to as vibrio, and corkscrew-shaped bacteria known as spirochaetes (Kotwal et al., 2022; Mishra & Chauhan, 2016; van Teeseling, de Pedro & Cava, 2017; Wahid, Ahmed & Habib, 2018; Yang, Blair & Salama, 2016).

Bacteria can be found everywhere in general; however, humans serve as ideal hosts for them (Kotwal et al., 2022). Every individual consists of normal flora, also referred to as nonpathogenic bacteria, within various body systems, particularly the digestive tract. These nonpathogenic bacteria protect humans from disease by competing with the nutrients pathogenic bacteria require to thrive and reproduce. Pathogenic bacteria can produce toxins that harm human tissues, act as parasites within human cells, or form colonies within the body, leading to disruptions in normal bodily functions (Fritz, Chaitow & Hymel, 2007). The dual nature of microorganisms highlights the importance of distinguishing between beneficial and harmful species. Identifying and classifying microorganisms, especially bacteria, is critical in various fields, including medical science, food safety, water quality monitoring, environmental surveillance, and biotechnology.

Therefore, the automated identification of bacteria is essential for a wide range of applications. The utilisation of automated methods for identifying and categorising bacteria in water sources plays a significant role in ensuring the safety of drinking water. Monitoring and maintaining water quality is crucial for safeguarding public health, as it allows for identifying hazardous microbes. The surveillance of bacteria in various environmental samples, such as air and soil, plays a vital role in comprehending ecological systems and identifying potential hazards, as demonstrated by environmental monitoring. The efficiency of this process can be enhanced through the utilisation of automated identification systems, which can also yield significant data for the purposes of environmental research and conservation.

In the food sector context, the expeditious identification of dangerous microorganisms in products is imperative, necessitating the implementation of automated bacterial identification systems. This process accelerates the retrieval of products from the market, preventing foodborne disease outbreaks. Implementing automated systems has the potential to enhance the efficacy of quality control protocols, consequently ensuring the safety of food items. The accurate and timely diagnosis of infectious diseases is of utmost importance in the medical industry, necessitating the use of automated methods for identifying microorganisms. This enables healthcare professionals to ascertain suitable interventions, oversee the progression of illnesses, and mitigate the spread of diseases. Automated technologies have the potential to greatly aid medical staff in the management and treatment of bacterial infections.

Hence, bacteria detection and classification are critical in medical science for the diagnosis of numerous diseases, treatment of infection, and trace-back of disease outbreaks. This is because it provides the basis for understanding many bacteria-contaminated diseases (Wahid, Ahmed & Habib, 2018). Accurate bacteria detection and classification can help identify and treat bacterial infections and monitor the effectiveness of antibiotic treatments, allowing appropriate treatment and management of the infection. Moreover, rapid and accurate detection and classification of bacteria that cause food poisoning, water contamination, and other public health concerns is critical for preventing the spread of disease and protecting public health. However, the diversity of bacteria in shapes, ranging from spherical to rod-shaped and spiral forms, increases the difficulties in performing bacteria classification tasks (van Teeseling, de Pedro & Cava, 2017; Yang, Blair & Salama, 2016). As a result, the classification of such microorganisms relies on factors such as cell structure, cellular metabolism, or differences in cell components (Wahid, Ahmed & Habib, 2018).

When the aetiology of an infection is unknown, broad-spectrum antibiotics are prescribed to patients as a first-line treatment. Unfortunately, the treatment’s efficacy is rarely sufficient. Moreover, it may impose a significant financial and psychological strain on the patient, as well as the potential risk of worsening the condition. On the other hand, with ongoing technological advancements, the volume of data gathered in microbial studies keeps increasing, making it a challenge to analyse and interpret the data using conventional methods, thus necessitating the use of more sophisticated computational techniques to extract the key features from the data landscape. However, there has been a noticeable decline in the number of taxonomists and classification experts in recent years. Additionally, traditional computational methods have been increasingly unmanageable when dealing with tremendous amounts of biological image data generated in microbiological research. This has created a pressing and critical demand for an automated method capable of rapidly and efficiently detecting and classifying bacteria with high sensitivity, high accuracy, and high precision to provide a robust alternative to the conventional methods of bacteria detection and classification that are time-consuming and require experts to read and quantify samples.

Automated bacteria detection and classification will serve as a vital element in improving the speed and accuracy of identifying and characterising bacteria. Deep learning (DL), a subset of machine learning (ML), is a rapidly advancing disruptive technology for developing artificial intelligence (AI) systems that can learn from large amounts of data. It employs deep neural networks to effectively model and solve complex problems (Zhang et al., 2022). These neural networks consist of multiple interconnected layers of “neurons” that process and transform input data to produce an output. DL has proven state-of-the-art performance in various biomedical applications (Krupa et al., 2022; Thakur et al., 2022). The superior performance of DL technologies in terms of accuracy and speed has shown their unprecedented potential in microscopy imaging for automatically detecting and classifying bacteria (Zhang et al., 2021). Both ML and DL methods can be highly accurate and efficient, and they can handle a large number of classes, but they require a large dataset of labelled examples to train the algorithms.

Object detection, a computer vision technique, is employed to detect and localise objects belonging to predetermined classes within digital images and videos in the form of bounding boxes with corresponding class labels and confidence levels for each detected object (Kaur & Singh, 2022). In the context of microscopy imaging, object detection enables the detection and classification of cells based on specific types or states. Such advancement can be integrated into smart imaging approaches, facilitating automated image acquisition processes (Spahn et al., 2022).

This work highlights the need to automate the detection and classification of bacteria, as conventional approaches, such as microscopy, are limited by their reliance on expert intervention and time-consuming processes. This study investigates different DL approaches, specifically the CNN object detection networks, to assess their efficacy in detecting bacterial cell cycle events, with a specific emphasis on E. coli bacterium. This study proposed an optimised You Only Look Only Once version 4 (YOLOv4) in detecting E. coli bacteria in microscopic images.

Research background

Bacteria consist of distinct cell shapes, ranging from spheres (cocci) to rods (bacilli) of different curvatures and helicities, and more exotic shapes, such as stars, moustaches, serpentines, spirals, and branches, representing a large, although undefined, proportion (van Teeseling, de Pedro & Cava, 2017; Yang, Blair & Salama, 2016). The morphology of the bacteria is influenced by a variety of activities, such as division or adaptations to local physical constraints (Garcia-Perez et al., 2021). Various approaches can be used for detecting and classifying bacteria, which can be categorised into conventional approaches and automated approaches. The detection and classification of bacteria are frequently done using microscopy, a common method of bacterial imaging.

Microscopy techniques are direct detection methods that utilise a microscope or an optical instrument to visualise the bacteria directly to obtain information about the bacteria’s morphology, such as shape and size, colonies, growth conditions, and other characteristics. The number of bacteria can be accurately measured because it is unnecessary to consider the amplification efficiency (Garcia-Perez et al., 2021). However, it requires highly trained and skilled personnel, and it might take time to pre-treat and observe the sample, but some bacteria have similarities in shape and size. Hence, computerising the process of bacteria detection and classification could reduce human effort and save valuable time via the implementation of ML and computer-vision technologies (Wahid, Ahmed & Habib, 2018).

ML is a subfield of AI that involves training algorithms to perform tasks without explicitly programming them. It relies on patterns and features in data to make predictions or take actions. It has a wide range of applications in various fields, including predictive modelling, recommendation systems, finance, marketing, computer vision such as facial recognition or object detection, natural language processing (NLP) like language translation or sentiment analysis, robotics such as object manipulation or navigation and healthcare such as medical diagnosis, medical image analysis, or patient care improvement (Sarker, 2021; Yusoff, Isa & Hasikin, 2013). An ML model typically has two phases: training and testing. In the training phase, samples are used as inputs of the ML model, and learning algorithms build the model from the features and patterns learned. In the testing phase, the learning model makes predictions for the previously unknown data using an execution engine.

ML-based automated bacteria detection and classification systems use ML algorithms that can learn from a dataset of labelled examples to classify new, unseen images of bacteria based on their features (Goodswen et al., 2021; Sarker, 2021). This can include techniques such as random forest (RF), support vector machine (SVM), k-nearest neighbour (k-NN), and decision trees (Goodswen et al., 2021; Kotwal et al., 2022). The steps involved in bacteria detection and classification using ML algorithms typically include the acquisition of bacterial images, preprocessing of acquired images, segmentation of images, feature extractions from images, and classification of bacteria (Kotwal et al., 2022). The study conducted by Ayas & Ekinci (2014) introduced a novel RF-based segmentation and classification approach for the automated classification of Mycobacterium tuberculosis (M. tuberculosis) bacteria in Ziehl-Neelsen (ZN)-stained sputum smear light-field microscopic images. The dataset of 116 images was collected from five different ZN-stained sputum smear slides taken from various patients, and the slides were prepared by the Mycobacteriology Laboratory at the Faculty of Medicine at Karadeniz Technical University. The dataset was divided into 40 images as the training dataset and 76 images as the testing dataset. The RF-supervised learning method was proposed to classify each pixel according to the local colour distributions as a part of candidate bacilli regions. The proposed method obtained 89.38% accuracy, 75.77% sensitivity, and 96.97% specificity.

Mohamed & Afify (2018) presented an ML algorithm for classifying ten-class bacteria species. The proposed system applied image preprocessing for enhancing image contrast and resizing images and employed the Bag-of-Words (BoW) model, which included Speeded Up Robust Features (SURF) descriptors for local image feature extraction, K-means clustering for generating the visual features and multiclass linear SVM as classifier. The experiment used the Digital Image of Bacterial Species (DIBaS) dataset created by the Chair of Microbiology of the Jagiellonian University in Krakow, Poland. The DIBaS dataset consisted of ten bacteria species with 20 images for each of them. The ten bacteria species were Acinetobacter baumannii (A. baumannii), Actinomyces israelii (A. israelii), Enterococcus faecium (E. faecium), Lactobacillus jensenii (L. jensenii), Lactobacillus paracasei (L. paracasei), Fusobacterium, Lactobacillus delbrueckii (L. delbrueckii), Lactobacillus reuteri (L. reuteri), Micrococcus species (spp.) and Candida albicans (C. albicans). The dataset is divided into 70% as the training dataset and 30% as the testing dataset. This proposed system achieved an average accuracy of 97% for classifying ten classes of bacteria, with eight classes of bacteria from the ten classes obtaining an accuracy of 100%.

An ML-based technique was implemented by Rahmayuna et al. (2018) for the pathogenic bacteria genus classification based on texture patterns from bacteria images. To enhance the image quality, the images were first preprocessed by implementing the contrast-limited adaptive histogram equalisation (CLAHE) method on the object images to generate an image with a clearer texture value and lesser noise. The features of preprocessed images were then extracted using texture analysis and Zernike Moment Invariant (ZMI) as an input for SVM. Two types of SVM kernels, including linear kernel and radial basis function (RBF) kernel, were then used to train and classify the selected features. The dataset obtained from the Kaggle website contained 600 optical images of foodborne pathogenic bacteria, with 150 images for each Escherichia species (sp.), Listeria sp., Salmonella sp., and Staphylococcus sp. The dataset was split into two groups with a ratio of 90:10, which are a training dataset of 540 images and a testing dataset of 60 images. The suggested model of SVM with RBF kernel showed slightly better overall performance as compared to the model of SVM with linear kernel. The SVM model with RBF kernel achieved a sensitivity of 0.9733, specificity of 0.9044, and accuracy of 90.33%, whereas The SVM model with linear kernel obtained a sensitivity of 0.9267, specificity of 0.9467 and accuracy of 89.83%.

Moreover, DL approaches could also be used to detect and classify bacteria. DL, a subset of ML, employs deep neural networks to effectively model and solve complex problems. These neural networks consist of multiple interconnected layers of “neurons” that process and transform input data to produce an output. DL has proven to be highly effective in a wide range of applications, including image and speech recognition, natural language processing, and gaming (Shukla & Muhuri, 2024; Traore, Kamsu-Foguem & Tangara, 2018). In the field of imaging, DL models can be used to analyse and interpret medical images such as X-rays, computerised tomography (CT), and single-photon emission computed tomography (SPECT) scans, as well as microscopy images (Mhathesh et al., 2021; Thakur et al., 2022). The most common type of DL algorithm is convolutional neural networks (CNNs), which effectively learn low-level features from image datasets, and they can be trained to classify bacteria from microscopic images based on their morphological characteristics (Kotwal et al., 2022; Mhathesh et al., 2021; Panicker et al., 2018; Spahn et al., 2022; Zhang et al., 2021; Zieliński et al., 2017). CNN provides significant results in image segmentation and classification and automation of feature classification from input images. It does not require manual feature selection for networks like traditional image processing and ML methods (Mhathesh et al., 2021).

A CNN technique that was combined with different colour spaces as input was presented by López et al. (2017) for identifying M. tuberculosis bacteria. The authors created a patch image dataset with 9,770 patches extracted from 492 extended depth-of-field (DOF) smear microscopy images dataset. The dataset was then augmented from 9770 patches to 29,310 patches by applying rotations at 90 and 180 degrees for each patch. Three CNN models were generated by composing the model with one, two, and three convolutional layers, respectively. Each CNN model was trained using RGB, R-G, and grayscale patch versions. This combination of CNN with different colour spaces allows the exploration of colour information of the M. tuberculosis bacteria. A receiver operating characteristic (ROC) analysis was implemented to compare the performance of CNN models with different input versions. The R-G input version of CNN models had shown the best results, with an accuracy of 98% and above. On the other hand, a robust, balanced fusion image dataset evaluated the CNN model with three convolution layers. This CNN model could classify positive and negative M. tuberculosis bacteria patches with an accuracy of 97% and above.

In the same year, Turra, Arrigoni & Signoroni (2017) trained a one-dimensional CNN (1D-CNN) model for the identification of hyperspectral urinary tract infection (UTI) pathogenic bacteria colonies. The 1D-CNN configuration possessed two convolutional layers, one pooling layer, one fully connected layer, and a final probability-based (softmax) layer for classification. A database of 16,642 bacteria colonies grown on the Petri dishes, including the colonies of E. coli, Enterococcus faecalis (E. faecalis), Staphylococcus aureus (S. aureus), Proteus mirabilis (P. mirabilis), Proteus vulgaris (P. vulgaris), Klebsiella pneumoniae (K. pneumoniae), Pseudomonas aeruginosa (Ps. aeruginosa) and Streptococcus agalactiae (Str. agalactiae), from 106 hyperspectral imaging (HSI) volumes was built by the authors and belonged to American Type Culture Collection (ATCC). The proposed CNN-based model, after 50,000 training iterations, was compared with SVM and RF, and it reached the best performance, showing an accuracy of 99.7%, whereas the SVM and RF achieved an accuracy of 99.5% and 93.8%, respectively.

The study conducted by Zieliński et al. (2017) introduced a deep CNN (DCNN) hybrid model for classifying bacteria genera and species. The DCNN method was applied for texture recognition to extract image descriptors, which were then encoded by the pooling encoder to produce a single feature vector and classified with SVM or RF. The experiment was based on the DIBaS dataset collected by the Chair of Microbiology of the Jagiellonian University in Krakow, Poland. The DIBaS dataset contained 660 images containing 33 different bacteria species. The developed model achieved an accuracy of the recognition of 97.24% for all of the classifiers, which were SVM and RF.

An automated M. tuberculosis bacteria detection and classification from microscopic sputum smear images using a DL-based algorithm was presented by Panicker et al. (2018). The suggested approach performed M. tuberculosis bacteria detection through image binarisation to denoise the input image for improving the image quality and subsequent pixel classification of the detected regions using CNN. The proposed CNN architecture contained three different layers and filters of convolution layers followed by a fully connected layer and, finally, a sigmoid output layer. The proposed system was evaluated using a dataset of 120 bacteria images with high-density and low-density backgrounds from cropping 900 positive patches and 900 negative patches. The 1800 patches were divided into 80% as a training dataset and 20% as a testing dataset. The presented algorithm achieved a sensitivity of 97.13%, precision of 78.4% and F-score of 86.76%.

An ‘Inception V1’ DCNN model was investigated by Wahid, Ahmed & Habib (2018) for the classification of both grayscale and RGB microscopy bacterial images. The approach consisted of several preprocessing steps for bacterial images, including manually cropping the images, converting them from grayscale to RGB, flipping them, and translating the resulting images. The experiment was conducted on the dataset of 500 microscopic images consisting of 5 different species of pathogenic bacteria, including Clostridium botulinum (C. botulinum), Vibrio cholerae (V. cholerae), Neisseria gonorrhoeae (N. gonorrhoeae), Borrelia burgdorferi (B. burgdorferi) and M. tuberculosis, collected from the online resources such as HOWMED, PIXNIO and Microbiology-in-Pictures. 80% of the images were used for training, and 20% of the images were used for testing. The ‘Inception V1’ DCNN model performed the classification of microscopic bacterial images and achieved a prediction accuracy of 95%.

Traore, Kamsu-Foguem & Tangara (2018) developed a DCNN-based model for image recognition of epidemic pathogenic bacteria, including V. cholerae and Plasmodium falciparum (P. falciparum), which caused the cholera epidemic diseases and malaria epidemic diseases, respectively, to automatically classify epidemic pathogenic bacteria from microscopic images. The proposed CNN architecture consisted of seven hidden layers, which are six convolution layers followed by a fully connected layer and a softmax layer, which was applied as the final classifier. The TensorFlow framework was employed to help implement the DCNN model for data automation, model tracking, performance monitoring, and model retraining. The dataset was acquired from Google images and contained 200 V. cholerae images and, 200 P. falciparum images for training, and 80 images for testing. The generated CNN model obtained an accuracy of 94%.

Mithra & Sam Emmanuel (2019) trained a deep belief neural networks (DBN) model for automatic detection and classification of M. tuberculosis bacteria present in the stained microscopic sputum images. The proposed system involved bacterial image preprocessing for noise reduction and intensity modification followed by image segmentation of the preprocessed bacterial images and extraction of intensity-based local bacilli features from segmented bacilli objects, and the extracted features were used as inputs for training the DBN algorithms for automatically classifying the M. tuberculosis bacteria and counting the number of bacteria present in the image to investigate the tuberculosis (TB) infection level. The sputum image dataset was from the Ziehl-Neelsen sputum smear microscopy image database (ZNSM-iDB). It contained 500 images, with 2,000 training objects, consisting of five categories: autofocus, no M. tuberculosis bacteria, few M. tuberculosis bacteria, overlapping and over-staining. Of the 500 images, 275 images of no, few and overlapping M. tuberculosis bacteria were used as training datasets and 225 images as testing datasets. The experiment results showed that the proposed system attained an accuracy of 97.55% with 97.86% sensitivity and 98.23% specificity.

El-Melegy, Mohamed & ElMelegy (2019) utilised the same dataset as the study conducted by Mithra & Sam Emmanuel (2019) to train a faster region-based convolutional neural network (R-CNN) for automated TB detection in images acquired with conventional bright field microscopy. The proposed approach implemented several data augmentation methods to increase the dataset size, including random image rotation, mirroring and random translations. 80% of the augmented images were randomly selected for training, and the residual images (20% of the augmented dataset) were used for testing. The faster R-CNN implemented in this study is a single, unified network for object detection, which combines a CNN, a region proposal network, a Region of Interest (RoI) Pooling layer, and a classifier. The authors utilised a pre-trained Visual Geometry Group (VGG)-16 net for the CNN part, which is a 16-layer CNN consisting of five convolutional layers, each followed by a max pooling layer, then three fully connected layers and a softmax layer. By using transfer learning, a pre-trained Faster R-CNN was trained for automated localisation and classification of M. tuberculosis bacteria. The experiment results showed that the proposed method achieved a precision of 82.6%, recall of 98.3% and F1-score of 89.7%.

In the same year, Treebupachatsakul & Poomrittigul (2019) developed a LeNet CNN architecture for the classification of two bacterial species in different cell shapes, S. aureus with spherical or round shaped and L. delbrueckii with long rod-shaped. The presented system used the Keras application programming interface (API) with the TensorFlow ML framework for bacterial species classification. The dataset of 800 digital images of S. aureus and L. delbrueckii that was created by the authors, with 80% of the images for training and 20% of the images for testing, was used to evaluate the performance of the proposed method and an accuracy of 75% was attained.

In 2020 Kang et al. (2020a) proposed a technique of hyperspectral microscope imaging (HMI) technology coupled with CNN, which acted as the classifier for classifying foodborne pathogenic bacterial species at the cellular level. HMI technology was used to provide spatial and spectral information on live bacterial cells. On the other hand, two CNN frameworks, U-Net and 1D-CNN, were implemented in the proposed system for automatic image segmentation using U-Net and spectral feature reduction and classification using 1D-CNN. The authors also used the other two principal components analysis (PCA)-based classifiers, PCA-KNN and PCA-SVM, for comparative purposes. For both PCA-KNN and PCA-SVM classifiers, PCA was used for spectral feature reduction, and then the KNN and SVM methods were used for classification. The dataset, which was acquired from the Poultry Microbiological Safety and Processing Research Unit (PMSPRU) of the U.S. Department of Agriculture located in Athens, Georgia, USA, consisted of 300 cells for each five foodborne bacteria species, including Campylobacter fetus (C. fetus), generic E. coli, Listeria innocua (L. innocua), Salmonella Typhimurium (S. Typhimurium), and S. aureus. 200 cells of each species were used as training dataset and 100 cells of each species were used as testing dataset. The proposed 1D-CNN model obtained an overall accuracy of 90%, which showed greater performance than the other two classification models, which were the PCA-KNN model, with an overall accuracy of 81%, and the PCA-VM model, with an overall accuracy of 81%.

Kang et al. (2020b) also introduced the Fusion-Net, a hybrid DL framework, to rapidly classify foodborne pathogenic bacteria at the single-cell level using HMI technology. HMI technology was adopted to detect foodborne bacteria and provide information on spatial (morphological features), spectral (spectral profiles), and combined spatial-spectral (intensity images). These features were analysed by employing three DL frameworks, including the long-short term memory (LSTM) network for processing the spatial features, deep residual neural network (ResNet) architecture for analysing the intensity images, and the 1D-CNN framework for analysing the spectral profiles. For Fusion-Net, the authors utilised a feature fusion strategy to stack these three DL frameworks to analyse morphological features, intensity images, and spectral profiles simultaneously. The authors obtained the dataset of five typical foodborne pathogenic bacterial cultures, including Campylobacter jejuni (C. jejuni), generic E. coli, L. innocua, S. aureus, and S. Typhimurium, from the PMSPRU of the USDA in Athens, Georgia, USA. The dataset of 5,000 bacterial cells was split into 72% for training, 18% for validation, and 10% for testing. The performance of Fusion-Net, with a classification accuracy of 98.4%, was superior to LSTM, ResNet, and 1D-CNN classifiers, with classification accuracies of 92.2%, 93.8%, and 96.2%, respectively.

Garcia-Perez et al. (2021) proposed a CNN-based technique for automatically detecting and classifying longitudinal division bacteria, Thiosymbion sp., based on phase contrast microscopic images. The proposed method included data preprocessing for extraction of samples and data labelling, followed by data splitting and data augmentation. The augmented data is then used as input to train the CNN model. The image dataset of 15,090 phase contrast microscopic images of Thiosymbion sp., with 2,244 and 12,846 images of the “longitudinal division” and “other division”, respectively, was partitioned into 52.8% for training, 13.2% for validation, and 33% for testing. By using transfer learning, a pre-trained CNN ResNet from the PyTorch framework was trained to automatically detect the longitudinal division of bacteria. The suggested method gained an accuracy of 99%.

A CNN-based technique for the classification of 3D light sheet fluorescence microscopy images of the gut bacteria in larval zebrafish, V. cholerae, was implemented by Mhathesh et al. (2021). The presented method was trained on the proposed CNN architecture Keras in-built plot function. The architecture consists of 11 layers containing an input layer, two convolution layers, two batch normalisation layers, two max-pooling layers, three dropouts, a flattened layer, and two dense layers. The suggested CNN system is trained on 3D light sheet fluorescence microscopy images of larval zebrafish, with the dataset of 21,000 manually labelled blobs for feature extraction, and achieved an accuracy of 95%.

Spahn et al. (2022) proposed a CNN-based approach, the YOLO version 2 (YOLOv2) algorithm, for E. coli cell cycle classification. The author contributed a dataset consisting of 40 bright field images of E. coli bacteria at three different growth stages, which are rod-shaped cells, dividing cells, and microcolonies. To obtain a well-trained model, the field of view size has to be selected so that the relative object size matches the networks’ receptive field. Hence, the authors split each image in the dataset into four regions, and the final training dataset contained 160 images, which were split into 100 images for training and 60 images for testing. The proposed approach involved data preprocessing steps of data annotation and data augmentation. The data augmentation techniques utilised were image rotation and flipping to increase the dataset size to four times its initial quantity. The proposed network gained a mean average precision (mAP) of 67%, precision of 73%, and recall of 74% in automatically detecting bacteria according to their corresponding growth stages.

Table 1 summarises the list of techniques from reviewed articles used for bacteria detection and classification using ML and DL approaches. Previous studies showed that both ML and DL methods can be highly accurate and efficient, and they can handle a large number of classes. While CNN-based models are widely used in object detection, not all CNN-based models, are specifically designed for this task, such as 1D-CNN (Kiranyaz et al., 2021; Krohling & Krohling, 2023; Singstad & Tronstad, 2020), Inception-V1 (Szegedy et al., 2015), LeNet (LeCun et al., 1998), FusionNet (Quan, Hildebrand & Jeong, 2021), and DBN (Hinton, 2009; Shukla & Muhuri, 2024). Although they can still be used in object detection pipelines, they are mostly used for other tasks like image classification, feature learning, or sequence data analysis. Furthermore, most detection and classification techniques involve tedious and complex processes such as image preprocessing and feature selection. This reinforces the necessity for automated approaches that can accelerate the detection and classification process.

Table 1 List of techniques from reviewed articles used for bacteria detection and classification using ML and DL approaches.

Method	Model	Data Preprocessing	Feature selection	Type of bacteria	Dataset	Performance evaluation	Authors	
Machine learning	RF	• Data annotation
• Image denoising
• Connected component analysis
• Image rotation
• Image resize
• Pixel segmentation	Pixel features	M. tuberculosis bacteria	116 ZN-stained sputum smear light-field microscopic images collected from 5 different slides taken from 5 patients	Acc = 67.98%
Se = 89.34%
Sp = 62.89%	Ayas & Ekinci (2014)	
	K-means clustering + SVM	• Grayscale conversion
• Histogram equalization
• Rescaling of image	SURF and LoH	Bacteria species	200 bacterial microscopic images obtained from the DIBaS dataset	Acc = 97%	Mohamed & Afify (2018)	
	SVM	• Image cropping
• CLAHE	ZMI and texture features	Pathogenic bacteria	600 optical images downloaded from the Kaggle website	Acc = 90.33%
Se = 97.33%
Sp = 90.44%	Rahmayuna et al. (2018)	
Deep learning	CNN-based	• Data augmentation
• Grayscale conversion and R-G conversion
• Data annotation	X	M. tuberculosis bacteria	Original dataset: 9,770 patches extracted from 492 extended depth-of-field smear microscopy images dataset
Augmented dataset: 29,310 patches	AUC = 99%
Acc = 99%	López et al. (2017)	
	1D-CNN	• Flat-field correcting and smoothing
• Image denoising
• Threshold-base foreground extraction and segmentation
• Cosine distance map weighting	Spatial-spectral features (intensity images)	UTI bacteria	16,642 bacteria colonies grown on the Petri dishes, from 106 HSI volumes	Acc = 99.7%	Turra, Arrigoni & Signoroni (2017)	
	CNN + SVM	X	Texture features	Bacteria colonies	660 microscopic images obtained from DIBaS dataset	Acc = 97.24%	Zieliński et al. (2017)	
	CNN-based	• Image denoising
• Image binarization
• Morphological opening and closing
• Data annotation	Pixel features	M. tuberculosis bacteria	1,800 patches extracted from 120 M. tuberculosis images with both high-density and low-density backgrounds	Se = 97.13%
P = 78.4%
F = 86.76%	Panicker et al. (2018)	
	Inception-V1 (GoogLeNet)	• Manual cropping
• RGB conversion
• Rescaling of image
• Random flipping
• Random translating	X	Pathogenic bacteria	500 digital microscopic images, collected from online resources such as HOWMED, PIXNIO and
Microbiology -in-Pictures	Acc = 95%	Wahid, Ahmed & Habib et al. (2018)	
	CNN-based	X	X	Epidemic pathogenic bacteria	400 digital microscopic images consisting of 200 Vibrio cholera images and 200 Plasmodiumfalciparum, downloaded from Google Images	Acc = 94%	Traore, Kamsu-Foguem & Tangara (2018)	
	DBN	• Adaptive median filtering
• Grayscale conversion
• Image segmentation	Intensity-based local bacilli features: LoH and SURF	M. tuberculosis bacteria	500 ZN-stained sputum smear microscopic images from online database, ZNSM-iDB	Acc = 97.55%
Se = 97.86%
Sp = 98.23%	Mithra & Sam Emmanuel (2019)	
	Faster R-CNN	• Data augmentation
• Data annotation	X	M. tuberculosis bacteria	500 ZN-stained sputum smear microscopic images from online database, ZNSM-iDB	P = 82.6%
R = 98.3%
F1 = 89.7%	El-Melegy, Mohamed & ElMelegy (2019)	
	LeNet	X	X	S. aureus and
L. delbrueckii	800 standard resolution bacteria images	Acc = 75%	Treebupachatsakul & Poomrittigul (2019)	
	1D-CNN	• Grayscale conversion
• Image segmentation	Morphological features and spectral profiles	Foodborne bacteria	300 cells from HMI images	Acc = 90%	Kang et al. (2020a)	
	FusionNet	• Grayscale conversion
• Min-max normalisation	Morphological features, spectral profiles, and intensity images	Foodborne bacterial cultures	5000 cells from HMI images	Acc = 98.4%
AUC = 1.0
TPR and FPR	Kang et al. (2020b)	
	ResNet	• Image binarisation
• Image segmentation
• Data augmentation
• Data annotation	X	Longitudinal division bacteria, Thiosymbion species	Original dataset: 730 phase contrast microscopic images, with 468 and 262 images of the “longitudinal division” and “other division” respectively
Augmented dataset: 15,090 phase contrast microscopic images, with 2,244 and 12,846 images of the “longitudinal division” and “other division” respectively	Acc = 99%
P, R and F1	Garcia-Perez et al. (2021)	
	CNN-based	• Image segmentation
• Data annotation	Blobs	Gut bacteria in larval zebrafish (V. cholerae)	21,000 blobs 3D light sheet fluorescence microscopy images	Acc = 95%	Mhathesh et al. (2021)	
	YOLOv2	• Data augmentation
• Data annotation	X	E. coli bacteria	Original dataset: 160 2D bright field microscopic images Augmented dataset: 800 2D bright field microscopic images	Acc = 67%
P = 73%
R = 74%	Spahn et al. (2022)	
Notes.

Acc, Accuracy; Se, Sensitivity; Sp, Specificity; P, Precision; R, Recall; F, F-score; F1, F1-score; AUC, Area under the curve; TPR, True-positive rate; FPR, False-positive rate.

The contribution and novelty of this work lie in addressing the challenges of bacteria detection and classification through the integration of DL techniques, particularly object detection networks, including SSD-MobileNetV2, EfficientDet, and YOLOv4. DL, an ML subfield, utilises neural networks to effectively represent intricate problem domains. As discussed, DL techniques, particularly CNN, have demonstrated potential in bacterial detection by leveraging morphological attributes. The study utilises transfer learning to train DL algorithms specifically tailored for object detection tasks in bacterial images, with a small-size dataset, fewer data preprocessing steps, and less computational effort compared to other existing studies in this field.

Materials & Methods

The proposed framework

The proposed framework of automated DL bacteria detection is presented in Fig. 1. This automated bacteria detection and classification systems were developed and tested on a few existing DL object detection algorithms, which were (i) Single Shot Detector-MobileNet version 2 (SSD-MobileNetV2), (ii) EfficientDet, and (iii) YOLOv4, and their performances were compared. As shown in the flow chart in Fig. 1, in this study, the overall processes of developing an automated bacteria detection and classification system involved data acquisition, data preprocessing that included data augmentation, data annotation and data splitting, development of object detection models and performance evaluation of object detection models.

Figure 1 The proposed object detection networks of deep learning algorithm for bacteria detection and classification based on their growth stages.

Data acquisition

The dataset employed in this study was collected from a publicly available repository, namely, Zenodo (https://zenodo.org/record/5551016#.Y9DwxHZBxPb), with the dataset title “DeepBacs –Escherichia coli growth stage object detection dataset and YOLOv2 model” contributed by Spahn et al. (2021).

Zenodo (https://zenodo.org/), launched in 2013, is an online, open-access repository built and operated by CERN and OpenAIRE which allows sharing of publications and supporting data (Hansson & Dahlgren, 2022; Peters et al., 2017; Sicilia, García-Barriocanal & Sánchez-Alonso, 2017). Zenodo provides researchers with an easy and stable platform to archive and publish their data and other output, such as software tools, manuals and project reports. The Zenodo repository was specifically designed to help ‘the long tail’ of researchers based at smaller institutions to share results in a wide variety of formats across all fields of science (Hansson & Dahlgren, 2022; Sicilia, García-Barriocanal & Sánchez-Alonso, 2017). Some communities are already using Zenodo in their archival workflows, and they are also taking benefits from their integration with the GitHub platform (https://github.com/) to easily preserve their GitHub repository in Zenodo. As Zenodo is intended to support individual researchers, it features no mechanisms to control the data uploaded (Sicilia, García-Barriocanal & Sánchez-Alonso, 2017). Zenodo has gained momentum and popularity, not only due to its integrated reporting lines for research funded by the European Commission but also due to the free assignment of Digital Object Identifier (DOI) to each publicly available upload to make the upload easily and uniquely citable and trackable (Peters et al., 2017; Sicilia, García-Barriocanal & Sánchez-Alonso, 2017). As Zenodo does not restrict the creation of communities by registered users, their creation and functioning respond only to the will of individuals and communities engaged with the repository. This makes the repository an interesting exemplar of a data curation repository in which researcher behaviour manifests both in the growth and actual use of the repository and also in the selection made by communities (Sicilia, García-Barriocanal & Sánchez-Alonso, 2017).

The collected dataset consists of 160 two-dimensional (2D) bright-field microscopic images of three growth stages of E. coli bacteria, including rod-shaped cells, dividing cells, and microcolonies, which are defined as 4+ cells in close contact. Each image consists of more than one cell and at least two growth stages of E. coli bacteria. An example of a dataset image consisting of the three growth stages of E. coli bacteria is shown in Fig. 2. The quantity of E. coli bacteria for each growth stage in the dataset was tabulated in Table 2. The 160 2D bright-field microscopic images comprise 1,097 rod-shaped E. coli cells, 697 dividing E. coli cells, and 157 E. coli microcolonies.

Figure 2 Dataset image consisting three growth stages of E. coli bacteria, including rod-shaped cell, dividing cell, and microcolony.

Table 2 Data distributions of E. coli bacteria for each growth stage in the public dataset.

Growth stages of E. coli cells	Before data augmentation	After data augmentation	
Rod-shaped E. coli cells	1,079	6,474	
Dividing E. coli cells	697	4,182	
Microcolony of E. coli cells	157	942	
Total	1,933	11,598	

Dataset preparation

Data augmentation

To reduce overfitting and enhance the performance of the DL model for object detection tasks, a substantial volume of data is essential for effective model training. Data augmentation is a technique employed to artificially expand the dataset size without actually collecting additional data. By applying a range of transformations to the existing data, data augmentation can effectively enhance the dataset’s size and diversity. These transformations can include random cropping, flipping, and rotation, among others. During the training process, the DL models treat the augmented images as distinct images, so they learn to recognise different variations of the same dataset images and, thus, achieve better performance.

The data augmentation was carried out with the aid of Augmentor, which is a Python library specifically designed for image augmentation in ML and DL applications (Bloice, Stocker & Holzinger, 2017). Augmentor automated the augmentation and artificial generation of image data to expand the dataset. The augmented dataset would then be implemented as the input of ML or DL algorithms. The dataset expansion involved employing various image augmentation approaches, including horizontal flip (fh) and vertical flip (fv) of the original image, as well as clockwise rotation of the original image by 90° (r90), 180° (r180), and 270° (r270). An example of the results of data augmentation of original images of E. coli cells is shown in Fig. 3, with a magnified region of the image shown in Fig. 4. The original dataset with 160 images, which consisted of 1,097 rod-shaped E. coli cells, 697 dividing E. coli cells and 157 E. coli microcolonies, was augmented to 960 images that consisted of 6,474 rod-shaped E. coli cells, 4,182 dividing E. coli cells and 942 E. coli microcolonies. The quantity of E. coli cells in each growth stage contained in the dataset before and after data augmentation is shown in Table 2. All augmented images were set at a resolution of 256 x 256 pixels, which was the same as the resolution of the original image.

Figure 3 Data augmentation of original image of E. coli bacteria.

Figure 4 Data augmentation of a magnified region of original image of E. coli bacteria.

Data annotation

Data annotation is a crucial process in the preprocessing stage when developing a DL model. It is a technique of identifying raw data in various formats, such as images, text files, video, etc., and including one or more meaningful and informative labels to provide contexts from which a DL model can learn. It is utilised to attribute, label, or tag the relevant information or metadata within a dataset so that a DL model can understand and recognise the input data and be trained precisely using DL algorithms.

Bounding boxes, which are basically rectangular boxes, are the annotation techniques most commonly used in computer vision for defining the location of the target object. These annotation boxes can be characterised by their x and y-axis coordinates in both the upper-left and lower-right corners of the rectangular box. In most cases, this annotation method is employed in object detection and localisation tasks.

The dataset images of E. coli bacteria were manually labelled with their corresponding growth stages by using the graphical image annotation software, namely LabelImg (Tzutalin, 2015). LabelImg was used to draw visual boxes in the image around each of the objects of interest, which were each E. coli bacteria in different growth stages. Each growth stage of E. coli bacteria that wanted to be detected was mandatory labelled in the dataset to avoid training issues. Additionally, only the visible part or visible and overlapped parts of the E. coli bacteria were labelled. The three growth stages of E. coli bacteria were represented by three classes as part of the configuration of LabelImg, as shown in Table 3. Three classes were configured for the bounding boxes, namely “Rod”, “Dividing”, and “Microcolony” for rod-shaped E. coli cells, dividing E. coli cells and E. coli microcolonies, respectively. An example of the annotation process is shown in Fig. 5. The annotations of each image were saved by LabelImg in XML files using the Pascal VOC format and in Text (TXT) files following the YOLO format. The Pascal VOC format is one of the universal image annotation formats for object detection algorithms such as SSD-MobileNetV2 and EfficientDet, whereas the YOLO format is a specific image annotation format for the YOLO algorithm for object detection. The Pascal VOC and YOLO annotation files will be carried forward for use in the following stage, the DL model training process, along with their respective image files.

Table 3 Image annotation process configuration.

Growth stages of E. coli cells	Class	Bounding box color	
Rod-shaped E. coli cells	0	Green	
Dividing E. coli cells	1	Yellow	
Microcolony of E. coli cells	2	Red	

Figure 5 Annotation process in LabelImg.

In Pascal VOC format, an XML annotation file was created for each of the images within the dataset. Each XML file had the identical name as its corresponding image, ensuring a clear association between the annotated data and its respective image. Each XML file consisted of the bounding box annotations, which are represented by an image folder, image filename, image path, image source, and image size with the description of width, height and depth, as well as an object with the description of name, ‘pose’, ‘truncated’, ‘difficult’ and bounding box (xmin, ymin, xmax and ymax). On the other hand, in YOLO format, each image in the dataset consisted of a corresponding TXT file sharing the identical name as the image, containing the bounding box annotations for that image, with object class, object coordinates, height, and width of the bounding box annotation of each single object in a single line.

Data splitting

The datasets in this study were split into two sets: training and testing. The training dataset was the subset of the original dataset used for training and building a DL model for object detection, whereas the testing dataset was used to measure the performance of the trained object detection models. In ML models, data splitting is one of the crucial elements that should be considered indispensable and highly necessary to eliminate or reduce bias in training data (Muraina, 2022). It can show how well the system learns and generalises to new data (Babu et al., 2024).

The selection of the split ratio selection should always be based on the size of the datasets. Many researchers suggested the 70:30 split ratio (70% for the training set and 30% for the testing set) as the best-split ratio, especially for a small number of the dataset size (Bhardwaj & Tiwari, 2015; Khorsheed & Al-Thubaity, 2013; Muraina, 2022; Nguyen et al., 2021; Pham et al., 2020). The studies conducted by Khorsheed & Al-Thubaity (2013), Bhardwaj & Tiwari (2015), Pham et al. (2020), and Nguyen et al. (2021) proved that 70/30 was the best training/testing ratio for getting the best performance of the AI models. Pham et al. (2020) revealed that the training set size generally had an important effect on the prediction ability of the AI models. Increasing the training dataset size improved the training performance and made the model more stable. For the testing performance, the increase in the training set’s size from 30% to 80% could also enhance the testing performance. However, when training size increased from 80% to 90%, the opposite trend was found in testing performance.

Therefore, in this study, the dataset was split with an optimal ratio of 70:30, which means that 70% of the dataset was allocated for model training, while the remaining 30% was allocated for model testing. The ratio of 70:30 allowed for an effective evaluation of the model’s performance on unseen data. Among the 960 images of the augmented dataset, 672 images were extracted to be used as the training dataset, whereas the remaining 288 images were used as the testing dataset. Table 4 shows the training and testing dataset before and after data augmentation.

Table 4 Training and testing datasets before and after data augmentation.

Dataset (percentage)	Quantity (images)	
	Original	After data augmentation	
Training (70%)	112	672	
Testing (30%)	48	288	
Total (100%)	160	960	

Development of object detection models

Once the dataset was augmented and labelled, it was then fed as input of the DL models for object detection. In this study, the proposed automated bacteria detection and classification systems were designed and tested with different object detection algorithms, including SSD-MobileNetV2, EfficientDet, and YOLOv4. Figure 6 summarises these DL models, which utilised CNNs as their base architecture. CNNs are highly effective for image analysis tasks, including object detection, due to their ability to learn and extract relevant features from input images. Additionally, transfer learning was utilised in this study. Transfer learning is an emerging DL approach successfully applied for various biomedical applications like breast cancer detection, cataract diagnosis, ECG classification, etc. The significance of the transfer learning approach is the ability to adapt the pre-trained model of one task to classify a distinct task without any concern about the size of the available dataset (Krupa et al., 2022).

Figure 6 Architecture of the proposed object detection models.

SSD-MobileNetV2

SSD-MobileNetV2 refers to a specific implementation of the SSD object detection algorithm that employs the MobileNetV2 architecture as its backbone network. As shown in Fig. 7, a combination of the SSD approach with MovbileNetV2 creates an architecture combining feature maps from multiple layers of the MobileNetV2 backbone network to directly predict bounding box coordinates and class probabilities without requiring an additional region proposal stage. It offers a good balance between detection accuracy and computational efficiency, allowing real-time object detection on devices with limited resources, like mobile phones and embedded systems.

Figure 7 SSD-MobileNetV2 architecture.

The SSD algorithm introduced by Liu et al. (2016) is a feed-forward convolutional network-based one-stage object detection framework. It operates by generating a fixed-size collection of bounding boxes and scores for the presence of object class instances in those boxes, followed by a non-maximum suppression step to refine and finalise the detections, resulting in accurate object localisation and classification. The overall network architecture of SSD, as shown in Fig. 8, is a modification of the MultiBox algorithm and bounding box proposals. SSD consists of two main components: a backbone network and an SSD head. The backbone network of SSD, which is truncated before any classification layers, can adopt various standard deep CNN architectures used for feature extraction from the input image, such as Visual Geometry Group Network (VGGNet), ResNet or MobileNet. The backbone network is then followed by the SSD head, which consists of several convolutional feature layers, which are multiscale feature map blocks and the final layer, or detection layer. These layers reduce in size from the previous layer and enable predictions of detections at various scales to achieve high detection accuracy with the use of relatively low-resolution input, further improving the detection speed. SSD generates different numbers of anchor boxes of varying sizes to detect objects of different sizes within an image by predicting the classes and offsets of these anchor boxes.

Figure 8 SSD architecture.

MobileNetV2 proposed by Sandler et al. (2018) is a CNN architecture that serves as an efficient and lightweight model based on TensorFlow for various computer vision tasks such as semantic segmentation, object detection and image classification. Notably, MobileNetV2 offers several advantages over other large-scale models, including low power consumption, reduced latency, and enhanced accuracy. Its design specifically caters to mobile and embedded vision applications, making it well-suited for resource-constrained environments. It is based on the inverted residual with linear bottleneck for a better balance between model size and accuracy. The linear bottlenecks, where the bottleneck layers employ a linear activation function, help in reducing information loss during feature extraction. A technique called depthwise-separable convolutions adopted by MobileNetV2 has great advantages in reducing the computational cost while retaining important spatial and channel-wise information. The bottleneck depthwise-separable convolutions with residuals are a drop-in replacement for standard convolutional layers, such as the backbone network of SSD architecture, VGGNet. The MobileNetV2 architecture starts with a full convolution layer comprising 32 filters, which is then followed by 19 residual bottleneck layers. As shown in Fig. 9, MobileNetV2 consists of three layers with two types of blocks. The first block is a residue block with stride 1, whereas the second block is a stride two layer used size reduction. The initial layer of MobileNetV2 is a 1 × 1 convolution layer incorporating non-linear activations like ReLU6. Subsequently, there is a depthwise convolution layer as the second layer, and finally, a 1 × 1 convolution layer with a linear activation function forms the third layer of MobileNetV2.

Figure 9 The main building block of MobileNetV2 architecture.

EfficientDet

EfficientDet, which was developed by Tan, Pang & Le (2020), is an advanced object detection model that prioritises a balanced compromise between model accuracy and computational efficiency. It builds upon the EfficientNet architecture for scalable and efficient object detection. Also, it incorporates several optimisation and backbone tweaks, such as using a weighted bi-directional feature network (BiFPN) for effective multi-scale feature fusion and a customised compound scaling approach for all backbone, feature network, and box/class prediction networks to be scaled uniformly for their resolution, depth and width simultaneously.

As shown in Fig. 10, EfficientDet consists of three major components: a backbone network, a feature network, and box/class prediction networks. It employs ImageNet-pre-trained EfficientNet (Tan & Le, 2019) as its backbone network that extracts multi-scale features from the input image. The BiFPN serves as the feature network, which takes multiple levels of features extracted by the backbone as input and outputs a set of fused features that represent the salient characteristics of the image. Lastly, a shared class/box prediction network utilises these fused features to predict the class and location of each detected object in the image. The BiFPN layers and class/box net layers are repeated multiple times to accommodate varying resource constraints and optimise the performance of the EfficientDet model.

Figure 10 EfficientDet architecture.

YOLOv4

YOLOv4 is a state-of-the-art high-precision and real-time object detection algorithm introduced by Bochkovskiy, Wang & Liao (2020), and built upon the previous YOLO series models. It is designed to strike the optimal balance between accuracy and speed, enabling real-time or near-real-time object detection on a variety of hardware platforms. It simultaneously predicts the coordinates of multiple bounding boxes with varying characteristics. These characteristics include classification results and confidence scores associated with each bounding box. Additionally, YOLOv4 dynamically adjusts the location of the predicted bounding boxes to improve the accuracy and precision of object localisation.

The overall network architecture of YOLOv4, as shown in Fig. 11, consists of a CSPDarknet53 backbone (Wang et al., 2020), a neck composed of Spatial Pyramid Pooling (SPP) (He et al., 2015) and Path Aggregation Network (PANet) (Liu et al., 2018) and a YOLOv3 head (Redmon & Farhadi, 2018). The CSPDarknet53 backbone extracts rich feature representations from the input image at different scales and levels of abstraction to capture both low-level and high-level features necessary for accurate object detection. The neck, composed of SPP and PAN, is used to collect feature maps from different levels or stages of the backbone network to enhance feature representation. The SPP additional module is inserted between CSPDarknet53 and PAN significantly to increase the receptive field to cover the increased size of the input network, separating out the most significant context features and minimising network operation speed reduction. PANet is used as the method of parameter aggregation from different backbone levels for different detector levels. YOLOv4 employs a YOLOv3 anchor-based head that consists of several detection layers responsible for predicting bounding box coordinates and class probabilities. These detection layers are typically built on top of the feature fusion output. YOLOv4 utilises anchor boxes of different scales and aspect ratios to capture objects of various shapes.

Figure 11 YOLOv4 architecture.

In addition, to improve the performance and capabilities of the YOLOv4, Bochkovskiy, Wang & Liao (2020) introduced Bag of Freebies (BoF) and Bag of Specials (BoS), which are the specific techniques and enhancements to be used by YOLOv4 for the backbone and detector. BoF refers to a set of techniques that are relatively easy to implement and improve the performance of the network without adding any extra computational overhead and sacrificing efficiency. The BoF techniques used in YOLOv4 include data augmentation, higher input resolution, improved optimiser and learning rate scheduling. On the other hand, BoS refers to a set of specialised techniques and modifications that significantly enhance performance but increase the computational complexity and inference time. The BoS techniques used in YOLOv4 include mist activation function, CSPDarknet53 backbone, SPP module and PANet.

Preprocessing and configuration

Prior to the start of training for SSD-MobileNetV2 and EfficientDet models, the training and testing datasets were converted from XML files to comma-separated values (CSV) files for training purposes. Then, a protocol buffer text (PBTXT) file and a TXT file of a label map with a list of classes of three growth stages of E. coli cells were created. Next, SSD-MobileNetV2 and EfficientDet models, as the TensorFlow Object Detection API frameworks (Yu et al., 2020), use datasets in TFRecord format for training; therefore, the images of both training and testing datasets were converted into the respective data file format called TFRecords, representing the dataset converted into binary format. In addition, the custom training configuration file for SSD-MobileNetV2 and EfficientDet models was created, respectively. The models were then trained and tested, and hyperparameters were further optimised, as summarised in Table 5.

Table 5 Hyperparameter settings for DL model training in detection and classification of E. coli cells depending on three growth stages (three classes).

Training hyperparameters	SSD-MobileNetV2	EfficientDet	YOLOv4	
Number of class	3	3	3	
Image size: Height	320	512	416	
Image size: Width	320	512	416	
Number of epochs	200	200	200	
Number of steps	12,000	48,000	–	
Number of iterations	–	–	6,000	
Batch size	16	4	32	
Subdivision	–	–	16	
Filter	–	64	24	
Learning rate	0.08	0.08	0.001	

Before starting the training process of the YOLOv4 model, the custom configuration file was created to configure Darknet by changing some training parameters in the configuration file provided in AlexeyAB’s GitHub (https://github.com/AlexeyAB/darknet). In the custom configuration file, the number of classes was defined as three to correspond to the three growth stages of E. coli bacteria that would be classified. The batch and subdivisions were set as 32 and 16, respectively, meaning that 32 images were loaded for one iteration, and the batch was split into 16 mini-batches. Each mini-batch consisted of 2 images that were processed together. This process would be repeated 16 times until the entire batch was processed, and subsequently, a new iteration began with 32 new images. The network size was set as 416, which is the value multiple of 32 for both width and height. Other training parameters in the configuration file, such as maximum batches (1), steps (2), and filters (3), were also changed for their values, as shown in Table 6, based on their respective formula provided in GitHub and the number of classes.

Table 6 Maximum batches, steps, and filters of the YOLOv4 model during training.

Training hyperparameter	Value	Note	
Maximum batches	6,000	The value of maximum batches should be not less than the number of training images and not less than 6,000.	
Steps	[4800, 5400]	–	
Filters	24	–	

(1) Maximum batches=number of classes×2000

(2) Steps=80%of maximum batches,90%of maximum batches

(3) Filters=number of classes+5×3.

Performance evaluation of object detection models

Evaluation of performance is essential for determining the precision of DL models used in object detection, such as identifying E. coli cells. This evaluation is based on parameters such as true positives (TP), which indicate correct identifications; false positives (FP), which occur when the model incorrectly detects objects that are not E. coli; and false negatives (FN), which occur when the model fails to detect E. coli. However, the concept of true negatives (TN) which is the accurate identification of non-E. coli objects is typically not used in this context, as the emphasis is on E. coli detection and not background elements.

Instead, the evaluation metrics generally focus on TPs, FPs, and FNs to provide insights into the performance and accuracy of the trained DL model in detecting and classifying E. coli bacteria according to the three growth stages. Mean average precision (mAP) is a specific performance metric commonly used for object detection tasks, such as detecting and localising objects within an image. It measures the precision–recall trade-off of the model across different confidence thresholds. Therefore, the evaluation of object detection algorithms primarily relies on the concepts of precision, recall, and F1-score, which provide a comprehensive assessment of the model’s performance. The precision, recall, and F1-score are defined as follows:

• Precision (4) is a metric that quantifies the accuracy of detections by measuring the ratio of TPs to the sum of TPs and FPs, meaning that precision represents the proportion of detected objects that are correctly identified (Godil et al., 2014). Precision measures the ability of a DL model trained for object detection to identify and classify only relevant objects. It is calculated as the percentage of correct positive predictions relative to the total number of positive predictions made by the DL model (Padilla, Netto & Da Silva, 2020). (4) Precision=TPTP+FP×100%

• Recall (5), which also refers to sensitivity, measures the ability of a DL model for object detection to correctly detect and capture all relevant objects. It is calculated as the number of TP relative to the sum of the TP and the FN. In other words, recall is the proportion of correctly detected items among all the items that should have been detected (Godil et al., 2014). It quantifies the percentage of correct positive predictions among all given ground truths (Padilla, Netto & Da Silva, 2020). (5) Recall=TPTP+FN×100%

• F1-score (6), often referred to as F-score or F-measure, is a commonly used metric that combines precision and recall into a single value. It provides a balanced assessment of a model’s performance by considering both the ability to correctly identify relevant items (precision) and the ability to capture all relevant items (recall). It is useful when there is an uneven class distribution or when both precision and recall are equally important. By taking into account both precision and recall obtained, the F1-score quantifies the overall accuracy of the model under test, reflecting the model’s overall performance in detection and classification tasks. It is computed as a weighted average of precision and recall. (6) F1−score=2⋅Precision×RecallPrecision+Recall

Results

Performance of object detection models on test images

The trained SSD-MobileNetV2, EfficientDet, and YOLOv4 models in detecting and classifying the E. coli cells from 2D bright-field microscopic images into rod-shaped E. coli cells, dividing E. coli cells, and E. coli microcolonies are tested by randomly selecting test images from the dataset, outputting images with bounding boxes which label the E. coli cells to their correct growth stage, and show the percentage of confidence level. The output images generated by the trained object detection models while testing are shown in Table 7, indicating the three proposed models accurately detect the rod-shaped E. coli cells, dividing E. coli cells, and E. coli microcolonies. The detected E. coli cells are surrounded in the bounding boxes, tagged with the classification of their respective growth stage, and provided with the confidence score of detection, which indicates the probability of the growth stages of E. coli bacteria being detected correctly.

Table 7 Performance results of the proposed models with examples of detected growth stages of E. coli bacteria.

Model	Output of test images	
SSD-MobileNetV2		
EfficientDet		
YOLOv4		

Based on the confidence score ranges from the results, the YOLOv4 model generally showed higher confidence scores, as compared to SSD-MobileNetV2 and EfficientDet models. Higher confidence scores in the trained YOLOv4 model imply that the model is more confident and certain in its detection of E. coli bacteria and their growth stages. This could be advantageous because the YOLOv4 model suggested a higher level of confidence in correctly detecting E. coli bacteria and their growth stages. Meanwhile, there are no E. coli bacteria are not being detected or wrongly detected for their growth stages by the trained YOLOv4 model. However, the SSD-MobileNetV2 and EfficientDet models miss detected (yellow arrow) or wrongly detected (yellow box) some of the E. coli bacteria.

On the other hand, the object detection models were required to detect multiple classes of growth stages of E. coli bacteria simultaneously. The proposed YOLOv4 model was able to create the different coloured bounding boxes for different classifications of the growth stages of the E. coli bacteria, whereas the proposed SSD-MobileNetV2 and EfficientDet were only able to create the same-coloured bounding boxes, green bounding boxes, for all classifications of the growth stages of the E. coli bacteria. In the YOLOv4 model, the rod-shaped E. coli cell, dividing E. coli cell, and E. coli microcolony were automatically assigned purple, orange and green bounding boxes, respectively. Associating specific colours to each class, each growth stage of E. coli bacteria in this case, provided clear labelling and annotation of the detected E. coli bacteria, helping visually distinguish and differentiate between E. coli bacteria of various growth stages. Therefore, the different coloured bounding boxes created by the YOLOv4 model in the output image for indication of different growth stages of detected E. coli bacteria supported multi-class detection. They allowed enhanced visual interpretation of the detected E. coli bacteria with their growth stages in the output image. Therefore, the user experience was improved as users can quickly identify and focus on specific objects or classes of interest while reviewing or analysing the detection results, resulting in more efficient analysis and decision-making.

Performance metric results of object detection models

The object detection models discussed in this section are multi-task bacterial image analysis using DL approaches as they allowed multi-class detection, with three classes of growth stages of E. coli bacteria to be categorised in this system, which were rod-shaped E. coli cell, dividing E. coli cell, E. coli microcolony. The test results of the test dataset were analysed statistically. The confidence threshold of each proposed object detection model was taken as 0.25 (or 25%), which was the minimum confidence level for an object detection to be considered valid. Thus, any detection with a confidence score below the 0.25 confidence threshold was considered a low-confidence detection and would be discarded. Meanwhile, the Intersection over Union (IoU) threshold of each proposed model was taken as 0.5 (or 50%). Hence, the detection was considered correct detection only if the IoU, which measures the ratio of the area of intersection between the predicted bounding box and the ground truth bounding box to the area of their union, exceeded the IoU threshold. The performance metrics of the proposed SSD-MobileNetV2, EfficientDet, and YOLOv4 models in detecting and classifying the growth stages of E. coli bacteria were evaluated, including precision, recall, F1-score, and mAP, and tabulated in Table 8.

Table 8 Summary of performance results of SSD-MobileNetV2, EfficientDet and YOLOv4 on detection and classification of E. coli bacteria into three growth stages.

Performance metrics	Proposed object detection model	
	SSD-MobileNetV2	EfficientDet	YOLOv4	
mAP (%)	96	97	98	
Precision (%)	71	72	86	
Recall (%)	78	77	97	
F1-score (%)	74	74	91	

Regarding the automated detection and classification of E. coli bacteria into three classes based on their growth stages, all of the three proposed object detection models had achieved a mAP of above 95%, with YOLOv4 achieving the highest mAP of 98%, followed by EfficientDet with 97% mAP, while SSD-MobileNetV2 achieving the lowest mAP of 96%. The mAP metric is the standard metric used for analysing the overall accuracy of an object detection model across multiple object classes. It measures the average value of average precision for each object class. Therefore, the highest mAP achieved by YOLOv4 indicated its superior performance in terms of overall detection accuracy in detecting and classifying objects as compared to the other two models. Although the EfficientDet and SSD-MobileNetV2 had lower mAP than that of YOLOv4, they still demonstrated good performance. Other than obtaining the highest mAP in this study, the trained YOLOv4 model also achieved the highest values for other performance metrics, including precision, recall, and F1-score, among the three proposed object detection models.

Furthermore, among the three proposed object detection models, the YOLOv4 model achieved the highest precision of 86% in detecting and classifying the growth stages of E. coli bacteria, indicating that 86% of positive detections made by the YOLOv4 model were correct and relevant, and the highest level of confidence in predictions. In addition, the EfficientDet and SSD-MobileNetV2 models both achieved an almost similar precision, with the EfficientDet obtaining a higher precision of 72% and the SSD-MobileNetV2 model attaining a lower precision of 71%. Therefore, the trained YOLOv4 model had the highest quality and reliability in the detection and classification of the growth stages of E. coli cells, followed by the EfficientDet and SSD-MobileNetV2 models.

In addition, out of all the three proposed object detection models, the YOLOv4 model also achieved the highest recall (sensitivity) of 97%, followed by the SSD-MobileNetV2 model with a recall of 78%, whereas the EfficietDet model attained the lowest recall of 77% which was just 1% lower than the recall of the SSD-MobileNetV2 model. A recall of 97% achieved by the YOLOv4 model showed that it correctly identified 97% of the true positive objects in the dataset. Hence, as compared to the EfficientDet and SSD-MobileNetV2 models, the YOLOv4 model had the lowest number of missed detections and the highest ability to correctly identify all true positive objects present in the dataset.

Moreover, the F1-score is a metric that indicates the harmonic mean of precision and recall, allowing access to the overall effectiveness of the three proposed DL models for object detection and providing a comprehensive evaluation of their performance. The YOLOv4 model achieved the highest F1-score of 91%, suggesting its excellent performance in terms of correctly detecting and classifying the E. coli bacteria while minimising incorrect detections. On the other hand, both EfficientDet and SSD-MobileNetV2 models obtained a lower and similar F1-score of 74%, and thus, they might have a higher probability of struggling with either missing relevant E. coli bacteria or producing a higher number of false detections compared to the YOLOv4 model.

Table 9 Performance achieved by the best proposed object detection model (YOLOv4) against the existing studies.

Source	DL algorithm	Type of bacteria	Database	mAP/Accuracy
(%)	Precision
(%)	Recall
(%)	F1-/F-score
(%)	
López et al. (2017)	CNN-based	M. tuberculosis
bacteria:
from:
• Bacterial images
• Non-bacterial
images	Self-prepared
dataset	99
(Accuracy)	–	–	–	
Turra, Arrigoni & Signoroni (2017)	1D-CNN	UTI bacteria:
•E. coli
•E. faecalis
•S. aureus
•P. mirabilis
•P. vulgaris
•K. pneumoniae
•Ps. aeruginosa
•Str. Agalactiae	Self-prepared
dataset	99.7
(Accuracy)	–	–	–	
Zieliński et al. (2017)	CNN + SVM	Bacteria
colonies	DIBaS database,
extended by the
authors	97.24
(Accuracy)	–	–	–	
Panicker et al. (2018)	CNN-based	M. tuberculosis
bacteria:
• Single bacilli
• Agglomerated
bacilli	TBImages Database,
contributed by
Costa et al. (2014)	–	78.4	97.13	86.76
(F-score)	
Wahid, Ahmed & Habib et al. (2018)	Inception-V1
(GoogLeNet)	Pathogenic
bacteria:
•C. botulinum
•V. cholerae
•N. gonorrhoeae
•B. burgdorferi
•M. tuberculosis	howMED, PIXNIO
and Microbiology-
in-Pictures	95
(Accuracy)	–	–	–	
Traore, Kamsu-Foguem & Tangara (2018)	CNN-based	Epidemic pathogenic
bacteria:
•V. cholerae
•P. falciparum	Google
Images	94
(Accuracy)	–	–	–	
Mithra & Sam Emmanuel (2019)	DBN	M. tuberculosis
bacteria:
• No bacillus
• Few bacillus
• Overlapping
bacillus	ZNSM-iDB,
contributed by
Shah et al. (2017)	97.55
(Accuracy)	–	97.86	–	
El-Melegy, Mohamed & ElMelegy (2019)	Faster R-CNN	M. tuberculosis
bacteria:
• Overlapping
bacillus	ZNSM-iDB	–	82.6	98.3	89.7
(F1-score)	
Treebupachatsakul & Poomrittigul (2019)	LeNet	•S. aureus
•L. delbrueckii	Self-prepared
dataset	75
(Accuracy)	–	–	–	
Kang et al. (2020a)	1D-CNN	Foodborne
bacteria:
•C. fetus
• generic E. coli
•L. innocua
•S. Typhimurium
•S. aureus	Self-prepared
dataset	90
(Accuracy)	–	–	–	
Kang et al. (2020b)	FusionNet	Foodborne bacterial
cultures:
•C. jejuni
• generic E. coli
•L. innocua
•S. aureus
•S. Typhimurium	Self-prepared
dataset	98.4
(Accuracy)	–	–	–	
Garcia-Perez et al. (2021)	ResNet	Longitudinal division
bacteria, Thiosymbion species:
• Longitudinal
division
• Other division	GitHub repository,
with title “Efficient
detection of longitudinal
bacteria fission using
transfer learning in
Deep Neural Networks”	99
(Accuracy)	–	–	–	
Mhathesh et al. (2021)	CNN-based	Gut bacteria in
larval zebrafish,
V. cholerae from:
• Bacterial images
• Non-bacterial
images	Self-prepared
dataset	95
(Accuracy)	–	–	–	
Spahn et al. (2022)	YOLOv2	E. coli bacteria
in different growth
stages:
• Rod-shaped cell
• Dividing cell
• Microcolony	Zenodo database,
with the title
“DeepBacs –
Escherichia coli
growth stage object
detection dataset
and YOLOv2 model”,
contributed by the
authors in their
previous study
(Spahn et al., 2021)	67
(mAP)	73	74	–	
Proposed object
detection model	YOLOv4	E. coli bacteria in
different growth
stages:
• Rod-shaped cell
• Dividing cell
• Microcolony	Zenodo database,
with the title
“DeepBacs –
Escherichia coli
growth stage object
detection dataset and
YOLOv2 model”, contributed
by Spahn et al. (2021)	98
(mAP)	86	97	91	

Table 9 presents a comparison of the performance achieved by the top-proposed object detection model, YOLOv4, against previous studies. Most of the previous research utilised different datasets and databases, making direct comparisons challenging. However, for a more meaningful comparison, this study focused on the study conducted by Spahn et al. (2022), as it employed the same dataset as this research. Spahn et al. (2022) also proposed an automated detection and classification system of E. coli bacteria according to the growth stages. The authors used the same original dataset as this study, which contains 160 2D bright-field microscopic images of three growth stages of E. coli bacteria before data augmentation. They partitioned the dataset into training and testing datasets with a ratio of 100:60, and the dataset size was increased 4x during training using data augmentation through image rotation and flipping. They implemented the DL approach developed by Redmon & Farhadi (2017), namely YOLOv2 or YOLO9000 network, with epochs of 100, batch size of 8, learning rate of 0.0003, and training time of 34 min. Their proposed system obtained an accuracy of 67% with 73% precision and 74% recall. The optimised YOLOv4 network, which is the best-proposed object detection model with mAP of 98%, precision of 86%, and recall of 97%, showed higher mAP, precision, and recall than the automated system proposed by Spahn et al. (2022).

The comparison of the training time and testing time of the proposed DL models in the detection and classification of E. coli bacteria based on their growth stages is shown in Fig. 12. The training time required for the YOLOv4 model, was the highest, which is 326 min, followed by the EfficientDet model with a training time of 235 min, whereas the SSD-MobileNetV2 model required the lowest training time of 60 min. Furthermore, YOLOv4, with 8 s of testing time, required the longest time for testing, followed by the EfficientDet model with a training time of 5.54 s, whereas the SSD-MobileNetV2 model required the lowest testing time of 5.3 s.

Figure 12 Training and testing time comparison of proposed DL models in detection and classification of growth stages of E. coli bacteria.

Discussion

This study has demonstrated the effectiveness and applicability of DL techniques, such as the SSD-MobileNetV2, EfficientDet, and YOLOv4 algorithms, in the context of bacterial image analysis. The proposed DL models, especially the optimised YOLOv4, which shows the best performance, can be applied in various research areas related to bacteria, specifically E. coli bacteria, for analysing bacterial microscopic images. The proposed YOLOv4 network offers reliable and consistent results with excellent accuracy and efficiency in detecting and classifying E. coli bacteria according to their growth stages, reducing the subjectivity and variability that can arise from the human interpretation and thus ensuring enhanced, more accurate and standardised bacterial analysis. This can support studies on E. coli bacterial behaviour, growth patterns, and characteristics and response to different environmental factors, which can be extrapolated to understanding other bacterial species, facilitating advancements in the field of microbiology.

In addition, by automating the process of detecting and classifying E. coli bacteria depending on their growth stage, the proposed YOLOv4 model contributes to significant time and cost savings compared to conventional methods, which involve manual techniques for detecting and classifying bacteria. This is because the proposed YOLOv4 model enables the rapid processing of large volumes of images, allowing for efficient analysis and reducing the reliance on manual analysis. Therefore, the availability of accurate and efficient DL models significantly improves efficiency in laboratory workflows, enabling researchers to focus on other aspects of their work, such as data interpretation and hypothesis testing or advanced research questions, and accelerate scientific discoveries.

The application of the proposed YOLOv4 algorithm in the analysis of bacterial microscopic images, specifically E. coli cells, can have significant implications in the biomedical field. Although the dataset used in this study consisted of non-harmful E. coli samples, the techniques and methodologies developed can be extended to the analysis of harmful bacterial strains as well, facilitating the identification and characterisation of pathogenic bacteria in clinical settings to aid in the diagnosis of E. coli infections. It can also support epidemiological studies, monitor antibiotic resistance, advance microbiological research, aid in environmental monitoring, particularly in water and food safety assessments, and improve laboratory efficiency. These applications contribute to the understanding, prevention, and management of E. coli-related or bacterial infections, ultimately improving human health and well-being.

This study demonstrated the capability of end-to-end ML for the automated identification of E. coli. The YOLOv4 model was optimised, and its performance was compared with other object detection algorithms. The contributions of this study are summarised as follows:

• The use of DL techniques, namely YOLOv4 object detection networks, signifies a fundamental transformation in the field of microbial identification. Conventional approaches, which include manual microscopy and the expertise of individuals for interpretation, are frequently characterised by their time-consuming nature and vulnerability to human error. The implementation of DL enables swift, automatic, and precise detection of bacteria, presenting a viable alternative across diverse applications.

• The proposed works address the obstacles associated with microbial detection, particularly in relation to bacteria, by incorporating deep-learning object detection networks. By placing emphasis on the activities occurring during the bacterial cell cycle, this work goes beyond mere identification and explores the intricate dynamic processes that take place inside microbial communities. The emphasis on E. coli serves as a tangible exemplification of the utilisation of DL in the realm of microbiological investigation.

• In addition, in the context of object detection, this study examines various DL techniques, including CNNs. This adaptability permits the adaptation of various models to specific imaging challenges involving microorganisms. The comparison of multiple open-source DL techniques provides valuable insights into their performance, thereby assisting researchers and practitioners in selecting the most appropriate approach for their particular requirements.

• Furthermore, by utilising transfer learning, this study successfully trained the high-accuracy DL algorithm specifically tailored for object detection tasks in bacterial images, with a small-size dataset, fewer data preprocessing steps and less computational effort as compared to other existing studies. Only two steps, data augmentation and data annotation, are used in data preprocessing. Training a deep learning model from scratch requires a large amount of data and computational resources. Therefore, by implementing transfer learning techniques, this study leveraged pre-trained models that have been trained on large-scale datasets for general object detection tasks as they have learned to recognise generic patterns and features in images. The pre-trained models were then fine-tuned on the specific dataset of bacterial images. This process involved tweaking the parameters of the pre-trained model so that it becomes more specialised in detecting and classifying bacterial features.

• The automated identification of bacterial cell cycle events, such as cell division, is indispensable for the advancement of microbiology research. Automation increases the pace of analysis and reduces reliance on human expertise, which is particularly relevant given the decline in taxonomists and classification specialists in recent years. With its capacity to learn from large datasets, DL provides a scalable solution for dealing with the growing volume of microbial data.

• Automated microbial detection has applications beyond the laboratory. Rapid and accurate identification of bacteria can greatly assist clinicians in determining appropriate treatments and averting the spread of disease during medical diagnosis. In addition, automated detection contributes to public health and safety measures in environmental monitoring, water quality assessment, and food safety.

Future works

The automated bacteria detection and classification system can advance the field of bacterial imaging analysis and contribute to improved diagnostics, research, and understanding of bacterial growth and behaviour. To address the limitations of the proposed solution and expand model generalisation, the proposed solution can be improved through a few strategies, such as dataset expansion, model optimisation, real-time deployment, and multimodal fusion.

The dataset of this study is relatively small even after data augmentation, with the imbalanced quantity of three growth stages of E. coli bacteria from bright-field microscopic images, which may lead to a higher risk of overfitting, where the DL model memorised the limited training examples instead of learning meaningful patterns. Meanwhile, different bacterial species exhibit variations in their morphological characteristics, such as shape, size, and structural features. Also, different bacterial imaging modalities used for producing bacterial microscopic images can introduce variations in image appearance and quality. Therefore, a larger and more diverse dataset of bacterial microscopic images that include different bacterial species and different microscopy imaging techniques should be collected and annotated to be implemented as the input of the DL model for detection and classification so that the DL model can learn to recognise the input data with various bacterial species in different bacterial microscopic images and be trained precisely, thus generalising better and improving their performance.

Besides, this study used transfer learning networks instead of developing a customised CNN with appropriate layers to match the detection and classification goals. Hence, further study can be conducted to optimise the best proposed DL model in this study. Architecture optimisation of the DL model involves exploring different backbone networks, modifying the existing architecture, and utilising advanced techniques such as model ensemble, whereas training hyperparameter optimisation can involve adjusting the model configurations. Model optimisation can help the models capture more informative features from bacterial microscopic images and improve their discriminative ability, thus improving their performance, efficiency, and generalisation capabilities in detecting and classifying bacteria.

Next, to facilitate the practical application of the automated bacteria detection and classification system, the trained DL model can be intergraded into a real-time automated bacterial image analysis system and developed into a user-friendly tool for researchers or microbiologists for applications where immediate bacteria detection and classification are required, such as in clinical settings or real-time monitoring of food and water safety. In addition, the usability and effectiveness of the automated bacteria and detection system can also be improved by integrating the automated system with real-time feedback mechanisms, enabling the user to interact with the system, validate results, provide feedback, and potentially enhance the performance of the automated system over time. Meanwhile, leveraging multimodal fusion techniques can be a valuable direction for future work. The fusion and integration of information from multiple modalities, such as incorporating the automated bacteria detection and classification system with additional sources that can provide gene expression data or allow biochemical measurements, obtain complementary information about bacterial species, virulence factors, or antibiotic resistance for bacterial analysis, thus improving the accuracy and robustness of bacterial detection and classification.

Conclusions

This study explores the effectiveness, potential, and applicability of DL approaches in bacterial image analysis, focusing on automating the detection and classification of bacteria from microscopic images. The proposed object detection networks of DL techniques, SSD-MobileNetV2, EfficientDet, and YOLOv4, have achieved promising results, with YOLOv4 outperforming the other models in detecting and classifying E. coli bacteria. YOLOv4 achieved the highest confidence score range without miss or wrong detection. Additionally, YOLOv4 could generate different coloured bounding boxes for different growth stages of E. coli bacteria, enhancing visual interpretation and allowing efficient and rapid review or analysis of detection results. Regarding statistical analysis, YOLOv4 achieves the highest mAP of 98%, precision of 86%, recall of 97%, and F1-score of 91%.

Additional Information and Declarations

Competing Interests

Author Contributions

Data Availability

The authors declare there are no competing interests.

Shuang Yee Chin conceived and designed the experiments, performed the experiments, analyzed the data, performed the computation work, prepared figures and/or tables, authored or reviewed drafts of the article, and approved the final draft.

Jian Dong analyzed the data, performed the computation work, prepared figures and/or tables, authored or reviewed drafts of the article, and approved the final draft.

Khairunnisa Hasikin conceived and designed the experiments, performed the experiments, authored or reviewed drafts of the article, and approved the final draft.

Romano Ngui conceived and designed the experiments, authored or reviewed drafts of the article, and approved the final draft.

Khin Wee Lai performed the computation work, authored or reviewed drafts of the article, and approved the final draft.

Pauline Shan Qing Yeoh performed the experiments, prepared figures and/or tables, and approved the final draft.

Xiang Wu conceived and designed the experiments, authored or reviewed drafts of the article, and approved the final draft.

The following information was supplied regarding data availability:

The code is available at GitHub: https://github.com/shuangyee9/Bacterial-Image-Analysis-using-Multi-Task-Deep-Learning-Approaches-for-Clinical-Microscopy.

The data is available at Zenodo: Spahn, C., & Heilemann, M. (2021). DeepBacs –Escherichia coli growth stage object detection dataset and YOLOv2 model [Data set]. Zenodo. https://doi.org/10.5281/zenodo.5551016.

The code and augmented dataset are available at Zenodo: Chin, S. Y., & Hasikin, K. (2024). Bacterial image analysis using multi-task deep learning approaches for clinical microscopy. Zenodo. https://doi.org/10.5281/zenodo.12625059.

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
