# Peer review of "Bacterial image analysis using multi-task deep learning approaches for clinical microscopy"

_PeerJ Computer Science, doi:10.7717/peerj-cs.2180_

## Round 0.1 · original submission · Major Revisions

Dear authors,

Thank you for submitting your article. Feedback from the reviewers is now available. It is not recommended that your article be published in its current format. However, we strongly recommend that you address the issues raised by the reviewers, especially those related to readability, experimental design and validity, and resubmit your paper after making the necessary changes.

Best wishes,

**Language Note:** The review process has identified that the English language must be improved. PeerJ can provide language editing services - please contact us at [email protected] for pricing (be sure to provide your manuscript number and title). Alternatively, you should make your own arrangements to improve the language quality and provide details in your response letter. – PeerJ Staff

Reviewer 1 ·

Basic reporting

In this context of my review, I make some suggestions to the authors and expect them to take them into consideration and edit them:
- The article contains some spelling and grammatical errors and should be corrected.
- Figure 6 is not clear, its resolution should be increased.
- An academic writing language should be used in the article. I believe that using personal pronouns (we) in the text negatively affects fluency.
- The YOLOv4 expansion does not need to be given everywhere it is used, it is sufficient to give it the first time it is used.
- The sentence "All augmented images were set at a resolution of 256 pixels x 256 pixels, which was the same as the resolution of the original image" should be edited. The first word "pixels" should be removed.
- Listing studies that use deep learning techniques for bacterial detection and classification is not sufficient for the literature review. Instead, relevant studies should be analyzed and the success of the studies should be given with performance metrics. After the relevant studies have been sufficiently examined, the contributions and novelty of the study should be presented.

Experimental design

- The organization of the introduction part of the study is weak. The flow has been disrupted by including the relevant studies section among the sections where the importance of bacterial detection and classification is emphasized. It would be appropriate to place the related studies section before the contributions and innovation section of the study.
- Why was the training and test dataset chosen as 70:30?
- The experimental results obtained were compared with only one study using the same data set. This comparison can be listed in a table or visualized graphically.
-Additionally, a table regarding studies developed using different methods and data sets for the relevant problem can be added to the experimental results section.

Validity of the findings

- The contribution and innovation of the paper are not clear enough. Innovation and contribution should be examined in detail based on relevant studies. How was the "addressing the challenges of bacterial detection and classification with deep learning methods" mentioned in the contribution achieved? It should be explained.

Additional comments

In this paper, the authors proposed a deep-learning approach based on SSD-MobileNetV2, EfficientDet, and YOLOv4 algorithms for bacterial image analysis. Thank you to the authors. The paper needs to be edited, taking into account the evaluations I have examined under the 3 sections above.

·

Basic reporting

The author has proposed a Bacterial image analysis using deep learning technique.
Author should provide the value of data partitioning for the proposed object detection networks along with justification.

Experimental design

The author should mention provide the proper literature review and justify why they have used object detection algorithms SSD-MobileNetV2, EûcientDet and YOLOv4.

Validity of the findings

Moreover, author should provide justification why they have used Zenodo repository. Furthermore, they can compare the results of various repositories with Zenodo.
Moreover, the author can compare the research work with more recent articles from reputed journals with their proposed work and prepare the comparative analysis table.

---

## Round 0.2 · accepted · Accept

Dear authors,

Thank you for the revision and for clearly addressing all the reviewers' comments. I confirm that the paper is improved. Your paper is now acceptable for publication in light of this revision.

Best wishes,

Reviewer 1 ·

Basic reporting

No comment.

Experimental design

No comment.

Validity of the findings

No comment.

Additional comments

The authors made the necessary edits to their papers in line with the suggestions. Therefore, I believe that publication is acceptable.